# Trametinib-Resistant Melanoma Cells Displaying MITF^high^/NGFR^low^/IL-8^low^ Phenotype Are Highly Responsive to Alternating Periods of Drug Withdrawal and Drug Rechallenge

**DOI:** 10.3390/ijms24097891

**Published:** 2023-04-26

**Authors:** Paulina Koziej, Katarzyna Kluszczynska, Mariusz L. Hartman, Malgorzata Czyz

**Affiliations:** Department of Molecular Biology of Cancer, Medical University of Lodz, 6/8 Mazowiecka Street, 92-215 Lodz, Poland

**Keywords:** cancer cell plasticity, drug holiday, drug rechallenge, IL-8, MITF, melanoma, NGFR, targeted therapy, trametinib resistance

## Abstract

Despite significant advances in targeted therapies against the hyperactivated BRAF^V600^/MEK pathway for patients with unresectable metastatic melanoma, acquired resistance remains an unsolved clinical problem. In this study, we focused on melanoma cells resistant to trametinib, an agent broadly used in combination therapies. Molecular and cellular changes were assessed during alternating periods of trametinib withdrawal and rechallenge in trametinib-resistant cell lines displaying either a differentiation phenotype (MITF^high^/NGFR^low^) or neural crest stem-like dedifferentiation phenotype (NGFR^high^/MITF^low^). Neither drug withdrawal nor drug rechallenge induced cell death, and instead of loss of fitness, trametinib-resistant melanoma cells adapted to altered conditions by phenotype switching. In resistant cells displaying a differentiation phenotype, trametinib withdrawal markedly decreased MITF level and activity, which was associated with reduced cell proliferation capacity, and induced stemness assessed as NGFR-positive cells and senescence features, including IL-8 expression and secretion. All these changes could be reversed by trametinib re-exposure, which emphasizes melanoma cell plasticity. Trametinib-resistant cells displaying a dedifferentiation phenotype were less responsive presumably due to the already low level of MITF, a master regulator of the melanoma phenotype. Considering new directions of the development of anti-melanoma treatment, our study suggests that the phenotype of melanomas resistant to targeted therapy might be a crucial determinant of the selection of second-line therapy for melanoma patients.

## 1. Introduction

Originating from melanocytes, melanoma is the most aggressive form of skin cancer. Several drugs targeting the hyperactivated B-RAF proto-oncogene (BRAF)^V600^/mitogen-activated protein kinase kinase (MEK)/extracellular signal-regulated kinase (ERK) pathway (vemurafenib, dabrafenib, and encorafenib against BRAF^V600^; trametinib, cobimetinib, and binimetinib against MEK1/2) have been approved for clinical use either alone or in combination. Unfortunately, resistance to BRAF^V600^/MEK1/2 inhibitors is widespread in melanoma, and tumor cells can adapt to these drugs along diverse mechanisms involving both additional genetic and non-genetic alterations [1,2,3,4,5,6,7,8,9]. Various genetic alterations, including BRAF^V600^ amplification and point mutations in genes encoding neuroblastoma Ras viral oncogene homolog (NRAS), Kirsten rat sarcoma viral proto-oncogene (KRAS), MEK1/2 and ERK1/2, loss of cyclin-dependent kinase inhibitor 2A (CDKN2A), and several other mutations, indicate that the genomic diversity can restrain the long-term efficacy of targeted therapy [5,6,9]. Genetic alterations cannot, however, fully explain clinical resistance to targeted therapies, and relapsed melanomas can exert noticeable transcriptomic alterations without a determined mutational background of resistance [2,10,11]. It is critically important to reduce death rates in melanoma patients who develop resistance to targeted therapy, and salvage therapies such as immunotherapy with inhibitors of cytotoxic T cell antigen 4 (CTLA-4) and programmed death 1 (PD-1) checkpoints are used in clinics [12]. Combination therapies that include inhibitors directed to additional targets are under development, but unacceptable treatment-associated toxicity is frequently an issue. Therefore, drug holiday, a strategy defined as intermittent dosing during therapy or temporal termination of drug administration after development of resistance, is a potentially attractive concept as it does not increase overall toxicity. It has been demonstrated in a melanoma preclinical model that intermittent dosing of vemurafenib, a BRAF^V600^ inhibitor, resulted in longer-lasting control of the tumor as compared with continuous drug administration [13]. Intermittent treatment of melanoma cells with encorafenib, another BRAF^V600^ inhibitor, delayed the development of resistance by adaptive re-sensitization to encorafenib rechallenge [14]. Contrarily, equal anticancer effects have been detected in intermittent and continuous treatments with either a BRAF^V600^ inhibitor or MEK inhibitor used alone or in combination in patient-derived xenografts [15]. In a randomized, open-label, phase II clinical trial (NCT02196181), intermittent dosing of BRAF^V600^ inhibitor dabrafenib and MEK inhibitor trametinib did not improve progression-free survival in melanoma patients with similar toxicity as in continuous dosing [16]. Similar results have been obtained recently for vemurafenib plus cobimetinib (NCT02583516) [17]. Most of these studies concern intermittent treatment prior the development of resistance to delay resistance onset. Preclinical models of acquired drug resistance to BRAF^V600^/MEK inhibitors are limited and do not provide consistent results presumably because of melanoma heterogeneity and different conditions applied to obtain and characterize drug-resistant cells that might affect the phenotype in different ways due to remarkable plasticity of melanoma [18,19,20,21,22,23,24,25]. A temporal discontinuation of treatment with BRAF^V600^/MEK inhibitors caused a drug addiction of cells, which resulted in the selective killing of drug-resistant cells, while remaining drug-sensitive cells could be eradicated by re-exposure to the drug [26]. In another preclinical study, however, drug discontinuation in melanoma xenografts that developed resistance to combined treatment with BRAF^V600^ and MEK inhibitors caused the rapid regrowth of tumors [18]. Therefore, a better understanding of mechanisms associated with drug holiday/rechallenge in the relation to the type of resistance is needed. It seems to be important to dissect and divide a drug resistance based on its biological determinants. In our previous studies, six drug-naïve melanoma cell lines derived from different patients, after being continuously treated for about 5 months with increasing concentrations of either vemurafenib or trametinib, generated eleven melanoma cell lines resistant to either of these drugs [8]. This mimicked to some extent the clinical situation, in which the acquired resistance occurs approximately six months after initial treatment. Those eleven resistant melanoma cell lines displayed no similar pattern of genetic and non-genetic alterations [8]. Whole-exome sequencing revealed novel resistance-associated genetic alterations but no additional mutations have been detected after a few months of drug-resistant cell culturing. Most of the genetic and non-genetic alterations were cell line-specific and drug-specific, even if they developed in the same drug-naïve cell line [8]. As targeted therapies against melanoma mainly consist of the combination of a BRAF^V600^ inhibitor and MEK inhibitor, but the development of resistance may apply to only one of these agents, we focus on resistance to trametinib, an inhibitor of MEK broadly used in anticancer therapy [27,28,29,30]. Considering the genetic and phenotypic heterogeneity of resistant melanomas, for the present study, we have chosen two stable trametinib-resistant (TRAR) BRAF^V600^ melanoma cell lines, 29_TRAR and 21_TRAR, representing entirely distinct phenotypes. While the 29_TRAR cell line maintained the melanocytic differentiation phenotype with high percentages of cells expressing the microphthalmia-associated transcription factor (MITF^high^) and low percentages of nerve growth factor receptor-positive cells (NGFR^low^) and senescence cells, the 21_TRAR cell line represented a neural crest stem-like dedifferentiation phenotype (MITF^low^/NGFR^high^) with a high percentage of senescence cells. We investigated alteration in phenotypes of these two trametinib-resistant melanoma cell lines in response to drug withdrawal (drug holiday) and drug rechallenge.

## 2. Results

### 2.1. Concept of the Study

Initially, three trametinib-resistant BRAF^V600^ melanoma cell lines, 21_TRAR, 28_TRAR, and 21_TRAR, generated from three different patient-derived cell lines, DMBC21, DMBC28, and DMBC29, respectively [7,8], were considered. These trametinib-resistant cell lines differed in phenotypes, including the percentages of MITF^high^ cells that were 14% ± 6.1%, 47% ± 5.5%, and 76.1% ± 9.9%, respectively [7], and the percentages of NGFR^high^ cells that were 53% ± 10.3%, 43.9% ± 3.2%, and 4.2% ± 0.2%, respectively [8]. Preliminary results indicated that they also differed in the percentages of senescent cells and secretion levels of IL-8 with the highest assessed for the 21_TRAR cell line and the lowest for the 29_TRAR cells (Figure 1A). The study was designed to compare changes at cellular and molecular levels induced by trametinib withdrawal named drug holiday (DH), followed by re-exposure to trametinib (re-TRA), and again drug holiday (re-DH) in two trametinib-resistant cell lines, displaying either differentiation phenotype (MITF^high^/NGFR^low^) or dedifferentiation phenotype (MITF^low^/NGFR^high^) as displayed in Figure 1B. Therefore, the two most representative cell lines, 29_TRAR and 21_TRAR, have been chosen for this long-term study assessing plasticity of trametinib-resistant cells enabling the transition between different cellular states, including differentiation, neural crest-like dedifferentiation, and senescence.

### 2.2. Reversible Phenotype Switching But Not Cell Death Is Observed in Trametinib-Resistant Melanoma Cells during Alternating Periods of Trametinib Withdrawal and Rechallenge

While massive cell death assessed by propidium iodide staining was not induced during drug withdrawal and drug rechallenge (Figure 2A), cell phenotypes were affected, and changes were distinct in the 29_TRAR and 21_TRAR cell lines. The proliferative capacity of the 29_TRAR cell line was reduced after drug withdrawal but stayed almost unchanged for the 21_TRAR cell line (Figure 2B). We have previously shown that the development of resistance to trametinib was associated with an increase in the percentages of cells expressing a marker of neural crest stem cells NGFR (CD271) only in selected drug-naïve cell lines, including the DMBC21 cell line but not in the DMBC29 cell line [8]. Interestingly, in the resistant 29_TRAR cell population, in which the percentage of NGFR-positive cells was very low (~3%), trametinib withdrawal induced a significant increase in the percentages of NGFR-positive cells (Figure 2C). This phenotype switch could be completely reversed by trametinib re-treatment. In the 21_TRAR cell line that already displayed a high percentage of NGFR-positive cells reaching 53%, no significant increase was detected after trametinib withdrawal (Figure 2C). To sum up, much more pronounced changes were observed in the 29_TRAR cell population than in the 21_TRAR cell population during alternating periods of trametinib withdrawal and rechallenge.

### 2.3. MITF Expression and Activity Is Reversibly Reduced by Trametinib Withdrawal in MITF^high^ Trametinib-Resistant Melanoma Cells

As MITF is a central regulator of melanoma phenotype switching, we analyzed MITF responsiveness to alternating periods of trametinib withdrawal and rechallenge. High levels of MITF and MITF-dependent DOPAchrome tautomerase (DCT), an enzyme crucial for melanosomal differentiation, were reduced to almost undetectable levels in trametinib-resistant 29_TRAR cells during the first and second rounds of drug holiday (Figure 3A). Expression of MITF and DCT was partially restored during re-exposure to trametinib, pointing at reversibility of the differentiation program. This was further confirmed by the assessment of changes in the expression of tyrosinase (TYR), another enzyme active in stage III/IV of melanin synthesis, whose expression is MITF-dependent (Figure 3B). Interestingly, substantial changes in MITF level and activity could already be detected one day after trametinib discontinuation (Figure 3C), suggesting that these changes are rather the consequences of cell reprogramming than the cell selection process. This is supported by the analysis of PI-positive cells showing that alterations in the cell phenotype induced by drug withdrawal/re-exposure were not accompanied by massive cell death (Figure 2A). In the trametinib-resistant 21_TRAR cell line, changes induced by drug holiday/re-exposure to trametinib were far less pronounced than those observed in the 29_TRAR cell line (Figure 3A–C). These discrepancies might be due to different original phenotypes of these two trametinib-resistant cell lines as shown in Figure 1A and Figure 2C. This is further supported by a comparison of expression levels of tyrosinase and DCT in these two resistant cell lines (Figure 3D). In general, the 29_TRAR cells that preserved the differentiation phenotype (MITF^high^) of their drug-naïve counterparts were more substantially affected by the trametinib withdrawal than the 21_TRAR cells (MITF^low^) that lost the differentiation phenotype.

### 2.4. Drug Holiday Induces Reversible Senescence in Trametinib-Resistant Melanoma Cells Exerting Differentiation Phenotype

Next, we explored whether changes in the MITF level and activity during alternating periods of trametinib withdrawal and rechallenge were accompanied by changes in other melanoma cell subpopulations. Using flow cytometry analysis, we observed cell enlargement and increased granularity induced by trametinib withdrawal in the 29_TRAR cell population (Figure 4A). We reasoned that drug holiday could conceivably induce senescence in those melanoma cells. To identify and quantify senescent cells, β-galactosidase staining was performed. The percentages of cells with senescence-associated β-galactosidase (SA-β-gal) activity were significantly raised in the 29_TRAR cell population on drug holiday, exceeding 50% of cells 10 days after trametinib withdrawal (Figure 4B). Interestingly, the process was partially reversed by re-exposure to trametinib, and again high percentages of senescent cells were re-established during the second round of drug holiday. As only small percentages of senescent cells (below 8%) were found in both the 29_TRAR and parental drug-naïve DMBC29 cell populations, this marked increase in percentages of senescent cells could be only assigned to the drug holiday. We further investigated whether the drug discontinuation-enhanced senescence in 29_TRAR cells could be associated with an elevated level of cyclin-dependent kinase inhibitor, p21^WAF1/Cip1^. Indeed, the p21 level was significantly raised in the first and second rounds of drug holiday when compared with its level in 29_TRAR cells exposed to trametinib, and this strong induction was reversible (Figure 4C). In the 21_TRAR cell line, senescent cells represented a large subpopulation and drug holiday did not additionally increase the percentage of senescent cells (Figure 4B). The level of p21^WAF1/Cip1^ in 21_TRAR cells was only increased with extended trametinib withdrawal (Figure 4C). Altogether, the drug holiday-induced senescence (DHIS) could be reverted by re-exposure to trametinib in trametinib-resistant melanoma cells displaying a high differentiation/low senescence phenotype. In trametinib-resistant cells showing a dedifferentiation/high senescence phenotype, DHIS was not induced.

### 2.5. IL-8 Expression and Secretion Are More Substantially Affected by Trametinib Cessation in Drug-Resistant Melanoma Cell Populations Exerting Differentiation Than Dedifferentiation Phenotype

We also investigated changes in the expression of interleukin-8 (IL-8, CXCL8) in trametinib-resistant cells after drug withdrawal and re-exposure to the drug. This cytokine is one of the most upregulated components of Senescence-Associated Secretory Phenotype (SASP) [19]. While the transcript level of IL-8 was significantly elevated in both trametinib-resistant cell lines after drug withdrawal, an increase in IL-8 expression was less pronounced in the 21_TRAR than in 29_TRAR cell line (Figure 5A). It might be due to the already higher level of IL-8 transcript in 21_TRAR cells than 29_TRAR cells prior to drug cessation (Figure 5B). Of note, a senescence state could be efficiently induced and reverted within a short time of 1–2 days, which indicates that a senescence program can be switched on and off in a large proportion of resistant melanoma cells but not in the rare isolated cells (Figure 4B and Figure 5A). We also assessed changes in the secretion of IL-8 by trametinib-resistant cells after drug withdrawal/rechallenge (Figure 5C). IL-8 secretion was upregulated in melanoma cells on drug holiday, again to a higher extent in 29_TRAR cell lines. The difference in IL-8 secretion between resistant cells growing without vs. with trametinib was significant for 29_TRAR cells but not for 21_TRAR cells. We also assessed the level of IL-8 secreted by 28_TRAR cells after drug withdrawal. The trametinib cessation induced only a minor increase in IL-8 secretion from 842 ± 25 pg/10^5^ cells per 1 mL by 28_TRAR cells (Figure 1A) to 893 pg/10^5^ cells per 1 mL and 899 pg/10^5^ cells per 1 mL by 28_TRAR cells one day and four days after drug discontinuation, respectively. This could again be partially explained by the markedly lower IL-8 secretion detected in a conditioned medium of 29_TRAR cells than 21_TRAR cells and 28_TRAR cells (Figure 1A). Thus, taking it all into account, IL-8 secretion was higher for resistant melanoma cell populations (21_TRAR, 21 DH, 28_TRAR, 28 DH, 29 DH) with a dedifferentiation phenotype (MITF^low^/NGFR^high^) than the 29_TRAR cell population grown in the presence of trametinib (TRAR) exerting a differentiation phenotype (MITF^high^/NGFR^low^).

## 3. Discussion

While the initial response to targeted therapies against the hyperactivated BRAF^V600^/MEK/ERK pathway brought hope for melanoma patients, acquired resistance is still an unsolved clinical problem. Phenotypic plasticity displayed by melanoma cells in response to fluctuating microenvironmental parameters results in various cell states such as differentiation/dedifferentiation, cancer stem-like state, senescence, cell dormancy, and quiescence, coexisting within a tumor and transiently generated [10,31,32]. Thus, it is difficult to fully overcome cancer resistance to treatment. In several preclinical studies performed using drug-naïve melanomas, various drug-tolerant subpopulations have been identified as a part of the response to BRAF^V600^/MEK inhibition [33], including a subpopulation of MITF^low^/NGFR^high^ cells [34,35,36]. While melanoma cell response to targeted therapeutics and their withdrawal after short exposure to drugs is a frequent subject of investigation (for example, [14,37]), reports on alterations induced by drug holiday and drug rechallenge in preclinical models of stable melanoma resistant to BRAF^V600^/MEK inhibitors [18,19,20,21,22,23,24,25] or in melanoma patients who developed resistance [38,39,40,41,42,43,44] are limited. In this long-term study, we focused on two distinct trametinib-resistant melanoma cell lines exerting either a differentiation phenotype (MITF^high^) or de-differentiation phenotype (MITF^low^).

First of all, neither drug withdrawal nor drug rechallenge induced extensive cell death in these two disparate trametinib-resistant BRAF^V600^-mutant melanoma cell lines as previously shown for BRAF^V600E^-mutant cell lines resistant to combined treatment with trametinib and dabrafenib after withdrawal of both drugs [45] and trametinib-resistant NRAS^Q61^-mutant cells after drug cessation [26]. Instead of loss of fitness, trametinib-resistant melanoma cells adapt to altered conditions by phenotype switching. Drug holiday-induced changes could be reversed by trametinib rechallenge, which emphasizes melanoma cell plasticity and phenotype dependency on microenvironmental conditions. The plasticity of trametinib-resistant cells could be easily detected as their phenotypes were altered within a short time. Trametinib withdrawal (1) reduced cell proliferation capacity and (2) induced stemness and (3) senescence features, including SASP. However, the extent of changes largely depended on the original phenotype of resistant melanoma cells, pointing to the importance of cancer heterogeneity, which is well recognized in clinics as differences between melanoma patients and how they respond to treatments. One of the possible explanations for discrepancies in response to alternating periods of trametinib withdrawal and rechallenge between investigated trametinib-resistant cell lines could be the level and activity of MITF, a main regulator of phenotype in melanoma [46,47,48]. In the constantly evolving MITF rheostat model, different levels of this transcription factor modulate distinct phenotypic states of melanoma cells with a high level supporting differentiation and proliferation, a lower level promoting invasiveness, and a very low level accompanying stemness and senescence [49,50,51,52,53]. MITF level-associated phenotype switching was evident in our study. In the MITF^high^ cell line (29_TRAR), trametinib withdrawal-triggered depletion of MITF was associated with reduced proliferation and differentiation, increased percentages of NGFR-positive stem-like cells, and enhanced senescence, whereas in the MITF^low^ cell line (21_TRAR), which was already proliferation^low^/differentiation^low^/NGFR^high^/senescence^high^, trametinib withdrawal only slightly reduced the MITF level and caused less-pronounced phenotypic changes. As a consequence, originally different trametinib-resistant cell lines became more similar at the phenotype level on drug holiday.

Two cell states, the neural crest stem cell-like state (NGFR^high^) and senescence that emerged in the context of MITF^low^ trametinib-resistance, could be detected either exclusively on drug holiday in the 29_TRAR DH cell population or regardless of the presence/absence of trametinib in the 21_TRAR cell population. The neural crest stem cell population disseminates during embryonic development into various cell lineages exerting different functions in the adult organism, including neuronal cells and melanocytes [54]. The re-emergence of the neural crest stem cell-like state (NGFR^high^) has been associated with diverse aspects of melanoma development and response to targeted therapeutics [35,55,56,57], resistance to various therapies [33,57], and immune evasion [58]. Interestingly, it has been demonstrated that the subpopulation of NGFR^high^ cells showed increased resistance not only to targeted therapies but also to adoptive T cell therapy and cytokines of activated T cells and natural killer (NK) cells such as interferon-gamma (IFNγ) and tumor necrosis factor (TNF) [58,59]. Knockdown of NGFR or pharmacological inhibition of NGFR were shown to contribute to T cell resensitization [58]. Sanchez-Del-Campo et al. demonstrated that the MITF^low^ melanoma cell population can be reduced by NK cell-mediated killing, whereas MITF^high^ cells can escape NK cell surveillance [60]. On the contrary, the most recently published study revealed that overexpression of NGFR in melanoma cells resulted in a reduction of NK cell infiltration into xenografts and NK cell-mediated melanoma cell killing [61]. The authors suggest that NGFR is a promising therapeutic target in melanoma and genetically engineered chimeric antigen receptor (CAR)-NGFR NK cells might be used against melanoma. In another report linking dedifferentiation of melanoma cells with acquired resistance to targeted therapy and inflammatory signaling from immunotherapy, the degree of dedifferentiation (upregulation of NGFR and downregulation of MITF) was associated with cell sensitivity to ferroptosis, suggesting that a ferroptosis-inducing drug could be a co-treatment component reducing the dedifferentiation-based resistance [47]. Our study showing that the NGFR^high^/MITF^low^ phenotype can either be generated already during the development of trametinib-resistance or reversibly after the cessation of trametinib if the resistant melanoma exerts a differentiation phenotype underlines the complexity of the non-genetic processes enabling melanoma switching within various phenotypic states. This should be considered in designing therapy for individual melanoma patients who developed resistance.

Another cell state that discriminates between these two examples of melanoma resistance is senescence that emerged already during the development of resistance (21_TRAR) or only after trametinib cessation in resistant cells but in a reversible manner (29_TRAR). Senescence of melanocytes contributes to skin aging [62], and BRAF^V600^-expressing melanocytes display features of senescence that could be silenced or reversed, leading to melanomagenesis [63,64]. Concentrations of circulating proteins of the Senescence-Associated Secretory Phenotype, including IL-8, are considered as candidate biomarkers of age and medical risk for cancer [65]. Growing evidence indicates that cell type and primary stressor are critical determinants in how the SASP can influence the development of cancer, whether it will be pro- or anti-tumorigenic [66]. It also becomes clear that anticancer therapy may induce the SASP, which in turn may impact the treatment efficacy [66]. While SASP factors secreted from cancer senescent cells can be initially cancer-suppressive [67,68,69,70], they can be mostly detrimental in the long term [71,72].

IL-8 expression in melanoma cells and its role in melanoma development, metastasis, and response to therapy have been the subject of several studies (reviewed in [73,74]). We previously showed that the IL-8 expression was significantly reduced in drug-naïve patient-derived melanoma cell lines by short exposure to trametinib or vemurafenib [35]. In melanoma patients, the serum IL-8 level was shown to correlate with tumor burden and objective response to BRAF^V600^ inhibitors vemurafenib and dabrafenib [75]. Changes in serum IL-8 levels also reflected the response of melanoma patients to anti-PD-1 treatment with either nivolumab or pembrolizumab [76]. IL-8 levels in serum were significantly reduced in therapy-responding patients and significantly increased during melanoma progression on treatment with anti-PD-1 therapy [76]. The elevated level of IL-8 has been identified as a predictive biomarker of reduced treatment benefit from immune checkpoint blockade in a large-scale retrospective analysis of a clinical study in patients with melanoma [77]. It has been suggested that IL-8 recruits immunosuppressive myeloid cells such as neutrophils to the tumor microenvironment to exclude T cells and/or their activation [77]. According to these reports, serum IL-8 levels could be used to predict melanoma response to targeted therapy and immunotherapy. While SASP proteins could be also released from non-transformed stroma cells to support tumor growth [78], the substantial contribution of melanoma cells to the production of IL-8 was evidenced in the experiment with surgery of melanoma xenografts, after which serum IL-8 level was reduced rapidly [75]. More importantly, the serum IL-8 level was found to be significantly decreased in patients after cancer-reduction surgery [75]. To our knowledge, there is no report analyzing IL-8 expression in melanoma cells resistant to targeted therapy and how this expression is affected by drug cessation. It would be of interest to further investigate whether IL-8 expression is mostly high in melanomas resistant to BRAF^600^/MEK inhibitors displaying the dedifferentiation phenotype, either shown originally by the resistant cells grown in the presence of a drug or induced by drug cessation in resistant melanomas that exert the differentiation phenotype. These questions are clinically relevant as anti-PD-1 and anti-CTLA-4 antibodies, used alone or in combination, are the second-line treatment option for BRAF^V600^ patients with melanoma resistant to targeted therapies [79]. Finally, to avoid ineffective second-line immunotherapy in melanoma patients with a high level of IL-8, it might be worth investigating whether immune checkpoint inhibitors can be combined with the agent(s) targeting IL-8 or its receptors to decrease immunosuppression within the tumor microenvironment. Anti-IL-8 therapies are already in clinical development in combination with immunotherapies for cancer patients, including patients with metastatic melanoma (NCT03161431; NCT03400332). The anti-IL-8 monoclonal antibody, BMS-986253, combined with nivolumab showed preliminary activity in melanoma progressing on checkpoint inhibitors, as reported in a phase I/II trial presented at ESMO Immuno-Oncology Congress 2022 [80]. None of these clinical trials, however, are designed for melanoma patients with acquired resistance to BRAF^600^/MEK inhibitors.

## 4. Materials and Methods

### 4.1. Compounds

Trametinib was purchased from Selleck Chemicals LLC (Houston, TX, USA) and used at 50 nM.

### 4.2. Cell Lines and Cultures

Drug-naïve melanoma cell lines were obtained from tumor specimens. The study was approved by the Ethical Commission of the Medical University of Lodz (RNN/84/09/KE). Informed consent was obtained from the patients. Cell lines were named DMBC21 and DMBC29 after the Department of Molecular Biology of Cancer. Trametinib-resistant cell lines (21_TRAR, 28_TRAR, and 29_TRAR) were obtained by continuous exposure of respective drug-naïve cell lines to increasing concentrations of trametinib, from 1 nM to 50 nM. Cells were cultured with or without trametinib at 50 nM in serum-free stem cell medium (SCM) composed of DMEM/F12 low osmolality medium (Gibco Thermo Fisher Scientific, Grand Island, NY, USA), B-27 supplement (Gibco, Paisley, UK), 10 μg/mL insulin, 1 ng/mL heparin, 10 ng/mL bFGF (basic fibroblast growth factor), 20 ng/mL EGF (epidermal growth factor) (BD Biosciences, San Jose, CA, USA), 100 IU/mL penicillin, 100 μg/mL streptomycin, and 2 µg/mL fungizone B. Cell cultures were maintained in low-adherent flasks (NUNC) at 37 °C in a humidified atmosphere containing 5% CO_2_. The medium was exchanged twice a week. A LookOut Mycoplasma qPCR Detection kit (Sigma-Aldrich, St. Louis, MO, USA) detecting multiple Mycoplasma and Acholeplasma species was utilized to test cell culture media, and the results were negative.

### 4.3. Propidium Iodide (PI) Staining and Flow Cytometry

Viability of melanoma cells was assessed before each experiment. Melanoma cell samples were collected, trypsinized, centrifuged, and stained with propidium iodide for 5 min at room temperature in the dark. Samples were measured by flow cytometer FACSVerse (BD Biosciences) and analyzed with BD FACSuite software.

### 4.4. Cell Confluency by Time-Lapse Fluorescence Microscopy (IncuCyte ZOOM)

Cells were seeded in 96-well plates (8 × 10^3^ viable cells per well). Changes in cell confluency were assessed as the area occupied by melanoma cells was monitored every 4 h using a time-lapse fluorescence microscope system (IncuCyte, Essen Bioscience). Quantification of the images was performed with the IncuCyte^®^ ZOOM basic analyzer (Essen, Bioscience).

### 4.5. NGFR-Positive Cells by Flow Cytometry

Flow cytometry was used to assess the percentages of NGFR-positive cells in the melanoma cell population grown with and without trametinib. To exclude dead cells from analysis, cells were incubated with a LIVE/DEAD fixable Violet Dead Cell Stain Kit (Life Technologies, Eugene, OR, USA) for 30 min at 4 °C in the dark followed by three washes and staining with PE-conjugated antibodies (anti-NGFR #557196, BD Biosciences) for 45 min at 4 °C in the dark. An appropriate PE-conjugated isotype control (#555749, BD Biosciences) was included in each experiment. Three washes were performed prior to analysis using flow cytometer FACSVerse (BD Biosciences). Data were processed by BD FACSuite software.

### 4.6. Cell Lysates and Western Blotting

Melanoma cells were lysed for 30 min at 4 °C in RIPA buffer consisting of 50 mM Tris pH = 8, 150 mM sodium chloride, 1% Triton X-100, 0.5% sodium deoxycholate, 0.1% sodium dodecyl sulfate, and a freshly added protease and phosphatase inhibitor cocktail (Sigma-Aldrich). After centrifugation, supernatants were collected and protein concentration was determined by Bradford assay (Biorad, Hercules, CA, USA) using a microplate reader Infinite M200Pro (Tecan Group Ltd., Salzburg, Austria) at 595 nm. Cell lysates were diluted in 2× Laemmli buffer consisting of 125 mM Tris pH = 6.8, 0.004% bromophenol blue, 20% glycerol, 4% sodium dodecyl sulfate, and 10% β-mercaptoethanol. Protein samples (15 μg) were separated on either 7% or 12% SDS-polyacrylamide gel. Electrophoresis was conducted at constant voltage of 25 V/cm. The proteins were transferred onto an Immobilon-PSQ membrane (Merck Millipore, Billerica, MA, USA) and Immobilon-P (Merck Millipore) from 12% and 7% gel, respectively. Nonspecific bindings were blocked with 5% non-fat milk in phosphate-buffered saline (PBS) containing 0.05% Tween-20 (Sigma-Aldrich) for 1 h. Primary antibodies against DCT and GAPDH (Santa Cruz Biotechnology, Santa Cruz, CA, USA) and MITF and p21 (Cell Signaling, Danvers, MA, USA) were used at a dilution of 1:1000 followed by binding of secondary HRP-linked anti-mouse or anti-rabbit antibodies (Cell Signaling) used at a dilution of 1:5000. After washing, the membrane was incubated with Clarity™ Western ECL Substrate (Bio-Rad, Hercules, CA, USA) and chemiluminescence was visualized using a ChemiDoc Imaging System (Bio-Rad). ImageJ software was used for quantification.

### 4.7. RNA Isolation, cDNA Synthesis, and Quantitative Real-Time PCR (qRT-PCR)

Total RNA was extracted and purified using the Total RNA Mini kit (A&A Biotechnology, Gdynia, Poland) according to the manufacturer’s protocol. RNA concentration was assessed using a NanoQuant Plate and a microplate reader Infinite M200Pro Tecan (Tecan) at 260 nm, and the purity of RNA samples was determined using a 260/280 nm ratio. An amount of 1 μg of total RNA was transcribed into cDNA using SuperScript II Reverse Transcriptase (Invitrogen Thermo Fisher Scientific, Carlsbad, CA, USA) and 300 ng of random primers. The quantitative real-time polymerase chain reaction was performed using a Rotor-Gene 3000 Real-Time DNA analysis system (Corbett Research, Mortlake, Australia). cDNA was amplified using a KAPA SYBR FAST qPCR Kit Universal 2X qPCR Master Mix (Sigma-Aldrich), 200 nM of each primer, and 25 ng of cDNA per reaction. The annealing temperature for all genes was 56 °C. Primer sequences were published previously for tyrosinase (TYR), ribosomal protein S17 (RPS17) [7], and IL-8 [8]. The relative abundance of each transcript was normalized to the level of RPS17 mRNA using a mathematical model with regard to the efficiency ratio.

### 4.8. Senescence Assay

Senescence was assessed as the percentage of senescence-associated β-galactosidase (SA-β-gal)-positive cells. For that, a Senescence Cells Histochemical Staining Kit (Sigma-Aldrich) was used according to the manufacturer’s protocol, as described previously [81]. After fixation, cells were incubated with a staining mixture for 18 h at 37 °C in an incubator without CO_2_ supplementation. Cell samples were subsequently transferred to microscope slides and observed under a microscope (Olympus BX41; Olympus Optical). At least 300 cells were counted to determine the percentages of SA-β-gal-positive cells.

### 4.9. Enzyme-Linked Immunosorbent Assay (ELISA)

To determine IL-8 secretion by melanoma cells to the culture medium, the ELISA kit Quantikine High Sensitivity Human CXCL8/IL-8 (HS800; R&D Systems, Minneapolis, MN, USA) was used according to the manufacturer’s instructions. In brief, cells (2.6–3.2 × 10^5^ cells per well) were plated in a 6-well plate for 24 h. Then, cell culture media were harvested and centrifuged at 17,978× *g* for 15 min at 4 °C. The supernatants were diluted 40×, added to each well of the plate coated with a monoclonal antibody specific for human IL-8, and incubated for 2 h. After six washes, the plate was incubated for 1 h with secondary antibodies conjugated to alkaline phosphatase, and another six washing steps were performed. The plate was incubated with the Substrate Solution, and the Amplifier Solution was added for color development. All steps were performed at room temperature. After 30 min, the reaction was stopped by adding sulfuric acid. The optical density of each well was immediately determined at 490 nm with wavelength correction at 650 nm using a microplate reader Infinite M200Pro (Tecan). The concentrations of IL-8 in the medium samples were calculated using a four-parameter logistic curve fit by comparing the optical density of the samples to the standard curve. The results are shown as pg of IL-8 per 10^5^ cells per 1 mL of the culture medium.

### 4.10. Statistical Analysis

Results are presented as mean values ± standard deviation (SD). They originate from at least three independent experiments unless otherwise indicated. The statistical significance of quantitative data was determined by a two-tailed unpaired Student’s *t*-test. Significance values were set at *p* ≤ 0.05 (*).

## 5. Conclusions

Despite great progress in the development of therapeutic strategies for advanced melanoma, much remains to be done before metastatic melanoma can be reduced to the chronic disease level. As the development of resistance to targeted therapeutics in melanoma cells is patient- and drug-specific, more effort is necessary to stratify melanoma patients for treatment strategies that could fit well to specific vulnerabilities arising with resistance. By providing insights into the remarkable plasticity of trametinib-resistant melanoma cells with a differentiation phenotype during alternating periods of trametinib withdrawal and rechallenge, this study highlights several issues that need to be addressed in clinics to receive improved responses in melanoma patients who developed resistance. In terms of the potential translational value of our study, the following may be considered: (1) reversibility of the phenotypic states that might influence the outcome of rechallenge with BRAF^V600^/MEK inhibitors as well as effectiveness of novel drug(s) applied either with targeted therapy or sequentially after targeted therapy discontinuation; and (2) the necessity to stratify melanoma patients for new treatment strategies according to specific resistance- or therapy discontinuation-associated phenotypes. However, future studies on melanoma specimens resistant to targeted therapies are needed to address the relevance of the phenotype in determining the next therapeutic strategy for melanomas resistant to BRAF^V600^/MEK inhibitors.

## Figures and Tables

**Figure 1 ijms-24-07891-f001:**
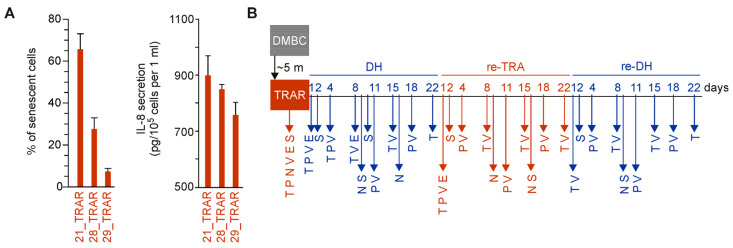
(**A**) Comparison of the percentages of SA-β-gal-positive cells and IL-8 secretion levels in 21_TRAR, 28_TRAR, and 29_TRAR cell lines. Mean ± S.D., n = 3 biological replicates. (**B**) The scheme presenting the experiment design. The molecular and cellular status of trametinib-resistant melanoma cells were assessed during alternating periods of drug withdrawal and drug rechallenge. Resistant cells were obtained from patient-derived drug-naïve cells (DMBC) after continuous exposure (~5 months) to increasing concentrations of trametinib. Resistant melanoma cells (TRAR) were subjected to trametinib withdrawal (DH), trametinib rechallenge (re-TRA), and the second round of drug holiday (re-DH). Biological samples were subjected to assessment of gene expression at the transcript (T) and protein (P) levels, viability (V), senescence (S), NGFR-positivity (N), and secretion of interleukin-8 (IL-8) (E) on indicated days. Trametinib-resistant cells (TRAR) exposed to the drug were assessed in each experiment and were used as a reference.

**Figure 2 ijms-24-07891-f002:**
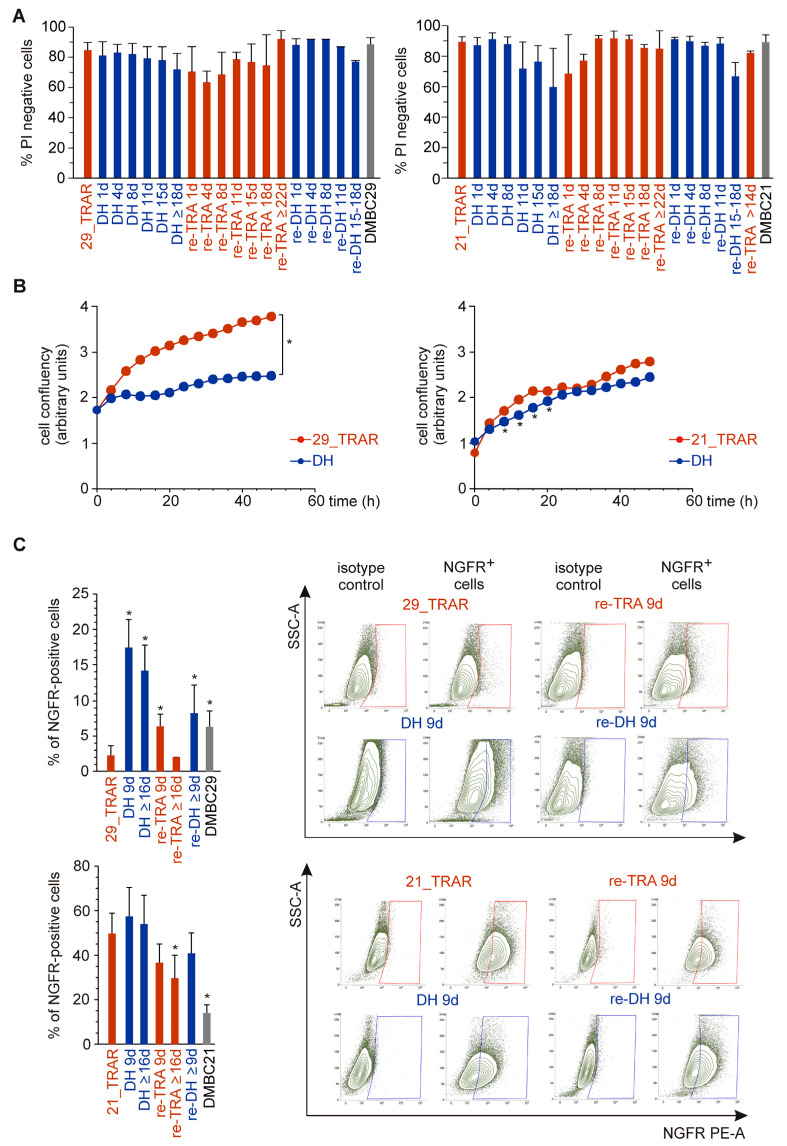
Phenotypes of trametinib-resistant melanoma cells displaying a differentiation phenotype (29_TRAR) and dedifferentiation phenotype (21_TRAR) are differently affected by drug withdrawal. (**A**) Cell viability across the intermittent treatment of trametinib-resistant melanoma cell lines (29_TRAR and 21_TRAR). Percentages of viable cells (propidium iodide (PI)-negative) were estimated on days of experiments assessing molecular and cellular alterations. Melanoma cells stained with PI were analyzed by flow cytometry. Mean ± S.D. (**B**) Resistant melanoma cells were grown with (TRAR) or without (DH) trametinib to assess changes in cell confluency monitored by time-lapse fluorescence microscopy (IncuCyte ZOOM). Results of a representative experiment performed in triplicate are shown. Statistically significant differences (*p* < 0.05) between data points for TRAR and DH are flagged with asterisks (*). The difference in cell confluency between 29_TRAR and 29 DH cells were statistically significant starting from x = 8 h. (**C**) The percentages of nerve growth factor receptor (NGFR)-positive cells were assessed by flow cytometry. Bars represent mean values ± S.D. of n = 3–7 biological replicates, *p* < 0.05 * Representative density plots are included.

**Figure 3 ijms-24-07891-f003:**
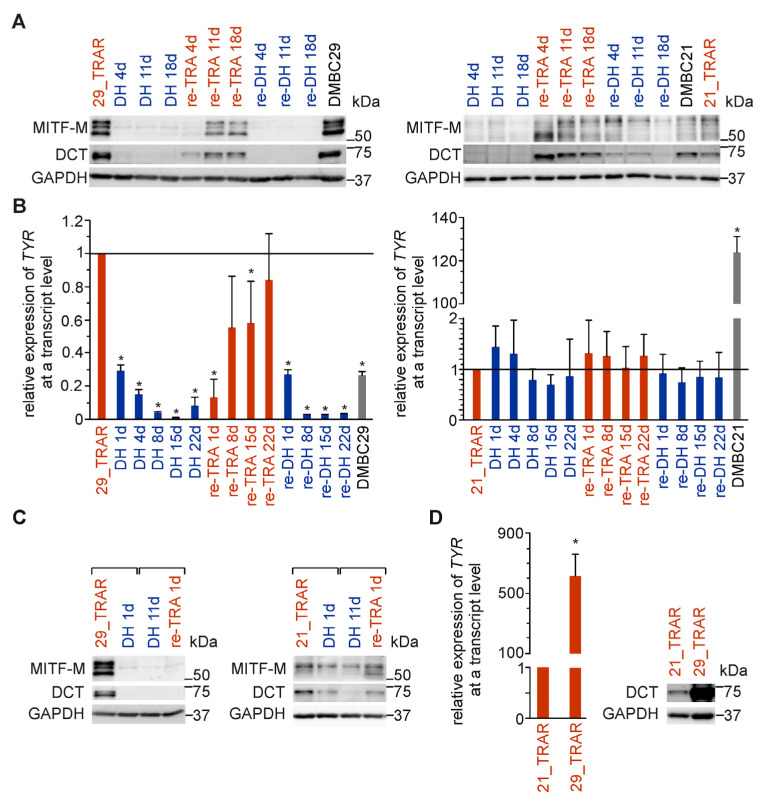
MITF level and activity in response to drug holiday and drug re-exposure in trametinib-resistant melanoma cells displaying a differentiation phenotype (MITF^high^) and dedifferentiation phenotype (MITF^low^). (**A**) The MITF level and its activity assessed as DCT expression. Whole cell lysates were prepared at different time points and immunoblotted with anti-MITF and anti-DCT antibodies. GAPDH was used as a loading control. Western blots are representative of three independent experiments. (**B**) Tyrosinase mRNA levels were assessed by qRT-PCR, normalized to the expression of ribosomal protein S17 (RPS17) and shown relative to the mRNA level in TRAR cells. Mean ± S.D., n = 3–4 biological replicates, *p* < 0.05 *. (**C**) Immediate response of MITF and DCT to trametinib withdrawal and rechallenge. Whole cell lysates were collected from TRAR cells, cells 1 day after trametinib withdrawal (DH 1d), and from cells 1 day after trametinib rechallenge (re-TRA 1d) preceded by drug holiday for 11 days (DH 11d). Western blots are representative of two independent experiments. (**D**) Expression of tyrosinase in trametinib-resistant cell lines, 29_TRAR vs. 21_TRAR, assessed by qRT-PCR and normalized to the expression of RPS17. n = 8–10. *p* < 0.05 * Comparison of DCT protein levels in 29_TRAR and 21_TRAR cells by immunoblotting.

**Figure 4 ijms-24-07891-f004:**
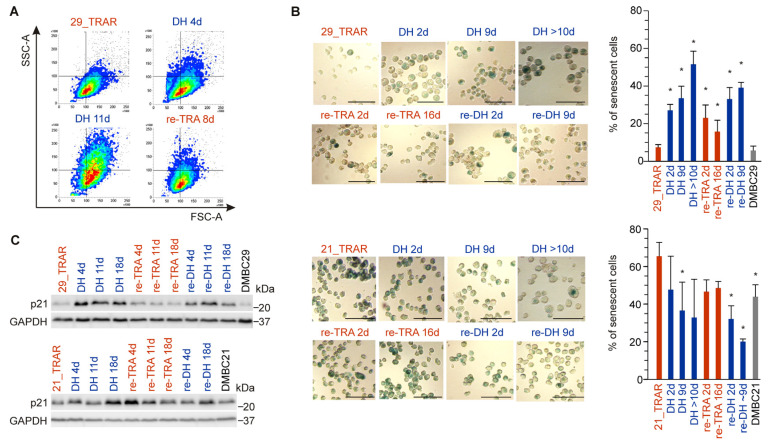
Distinct changes in senescence state induced by drug holiday (DH, re-DH) and re-exposure to the drug (re-TRA) in trametinib-resistant melanoma cell lines. (**A**) Assessment of cell size and granularity by the flow cytometry. (**B**) Representative images of SA-β-gal staining (bars = 50 μm) along with their quantification. The percentages of SA-β-gal-positive cells with respect to the total number of cells. Mean ± S.D., n = 3–4 biological replicates, *p* < 0.05 *. (**C**) p21 level evaluated by Western blotting. GAPDH was used as a loading control. Images are representative of two (re-DH) or three (other conditions) independent experiments.

**Figure 5 ijms-24-07891-f005:**
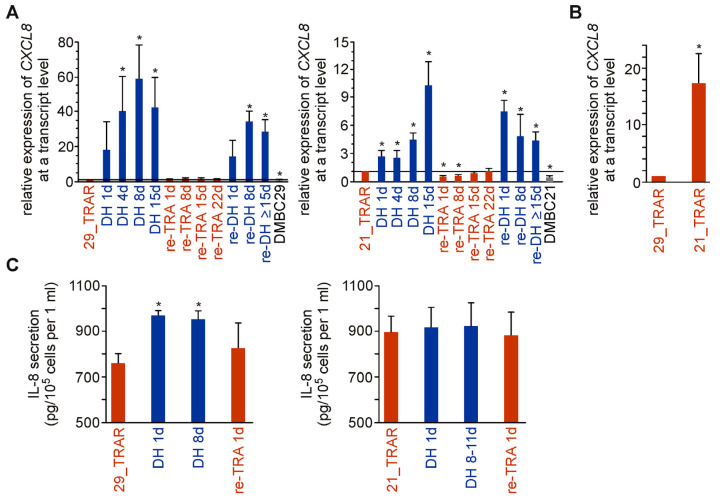
IL-8 expression and secretion are diversely affected by trametinib withdrawal in drug-resistant melanoma cells. (**A**) IL-8 mRNA levels determined by qRT-PCR and normalized to the expression of RPS17. Mean ± S.D. of n = 3–4 biological replicates for TRAR and DH, n = 2–3 biological replicates for re-TRA and re-DH, *p* < 0.05 *. (**B**) Transcript level of IL-8 in trametinib-resistant cell lines, 21_TRAR vs. 29_TRAR, assessed by qRT-PCR and normalized to the expression of RPS17. n = 19. *p* < 0.05 *. (**C**) Secretion of IL-8 assessed by ELISA in cell supernatants after 24 h of cell culture. Mean ± S.D. of n = 3. *p* < 0.05 *.

## Data Availability

All data presented in the study are available upon request from the corresponding author.

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
