# Peer review of "Trametinib-Resistant Melanoma Cells Displaying MITF^high^/NGFR^low^/IL-8^low^ Phenotype Are Highly Responsive to Alternating Periods of Drug Withdrawal and Drug Rechallenge"

_ijms, 2023, doi:10.3390/ijms24097891_

Round 1

Reviewer 1 Report

Dear Authors,

I think it is of an author's best interest to have a review with the highest amount of fair-criticism as possible, thus having his/her name associated with high-quality work. Minding the time constraints to review this paper, I spent the maximum amount of time I could on it and tried to be as critical as I could.

As a Reviewer my general opinion is that the topic is interesting.

However, I think the manuscript is not ready yet for publication, it still needs maturing in terms of the accuracy in theory and also in formatting.

Next, see my complete review, minor and major issues are blended. The comments follow the same order as the paper.

1) I think that your Abstract is well-written. But you should be more specific regarding the methods and the implementations which are tested and proposed in your work. As I can observe, you provide more general information in abstract, please be more specific. Are the methods new in the domain or new in the research? Thank you.

2) Please provide extra information related to previous works and design methodologies. Please extend your introduction and provide more state-of the art information. It will increase the quality of your work and you will receive easier citations. Moreover, add a Reminder of your work. Please also be careful and check all abreviations!

3) Regarding the Background, please explain more the necessity of this methodology. Please provide application example and real-world implementations. Also, the limitations of each methodology are not clearly provided.

4) Please explain more in Introduction the [1-8] references. What is their main sybject and their main contribution?

5) Please provide an extra Section, from example as Section 2 in which you refer to the design methology and the main steps.

6) Please provide first the Materials and Methods and afer that the Results. It will upgrade the design methology.

7) Please provide a general diagram (Figure) which provide your methodology in a more clear way (block diagram). If a reader see a figure will have a more clear opinion about your work.

8) Please provide more information regarding each sub-section of the Section 4. It will help the reader to understand more.

9) Please provide a Table in which you compare your work with previous related works. It will upgrade your work. (related metrics).

This review, is provided to the authors in order to upgrade the quality of this work. I hope to see your work published in this Journal as soon as possible.

Reviewer 2 Report

Vertically targeting both BRAF and MEK in MAPK-hyperactivated metastatic melanoma has revolutionized how those patients are treated for their cancer. However, acquired resistance to these dual inhibition strategies remains a major clinical challenge. Herein, Koziej et al describe results from studies with two trametinib-resistant cell lines that exhibit different differentiation states. Melanomas can exist along a spectrum of melanocyte/neural crest stem cell developmental spectrum as melanomas originate from melanocyte or melanocyte precursors found within the neural crest lineage. Koziej et al suggests that in one of these trametinib-resistant cell lines, phenotypic plasticity related to the developmental state of the line was altered following trametinib withdrawal or application. The reviewer applauds the efforts of the laboratory for using the matched parental and trametinib-resistant cells generated from a melanoma tumor that had acquired trametinib resistance. However, the description of the data described in the manuscript is superficial and mainly accounts to reporting a phenomenon associated with the 29_TRAR cell line. The changes seen with trametinib withdrawal and re-application frequently do not mirror the baseline levels of the differentiation marks of the 21_TRAR cell line, which is the differentiation state that the 29_TRAR cell is purportedly phenotype-switching to become more like. The authors state that a bank of matched pre- and post-resistant cell lines have been derived (see lines 48-52) and are described in a previous study. Expanding the study to examine these phenotypic plasticity claims in the full bank of cells to see more than one line matches the 29_TRAR results would be prudent. Specific comments about the manuscript are shown below:

1)    Figure 2C – If the cells are truly phenotypically plastic due to withdrawal or re-challenge of trametinib in the TRAR cells, why did MITF-M and DCT not increase rapidly with trametinib re-application in the 29_TRAR cells? The pattern I am describing is occurring in the 21_TRAR cells in the second 2C sub-panel. I am just unclear if phenotype switching is the right phrase for the phenomenon observed.

2)    Figure 1D – As mentioned above in the formatter, I am not sure what the biological significance of the phenotype switch is if the switch is not approaching the values seen at baseline with the 21_TRAR cell line? The same trends are at least seen between the two lines. Again, finding a second line that mimics 29_TRAR to show a more robust phenotypically plastic phenotype across multiple lines from multiple patients would be more compelling and convincing evidence.

3)    Figure 4 – Again, the changes observed with 29_TRAR at the mRNA level do not seem to equal what is seen with the baseline levels of IL-8 mRNA in 21_TRAR cells which makes me question whether these changes are due to differentiation state switching. If IL-8 levels are the same regardless of differentiation state (based on these two lines), then this would rule out the Discussion point highlighted in lines 345-347.

4)    Figure 2A – What is the correct MITF-M band in the immunoblot? There appears to be 3 MITF-M bands. Also, do the authors have a mechanistic hypothesis as to why MITF is lowered following trametinib withdrawal when MITF levels are similar across the DMBC29 parental line and 29_TRAR cell lines? Lastly, in the 21_TRAR subpanel, please depict a blot that looks more like what is submitted in the raw blot files. Cropping over 21_TRAR to the left end of the blot is unnecessary and could be confusing to the reader (i.e., did these blots come from different blots). By using the image “as is” from the raw files, you would be helping the reader interpret these blots in a clearer manner.

5)    Figure 3 – I am not sure if the changes in senescence following trametinib withdrawal or reapplication can be attributed to differentiation state. These results are just correlative with other observations made throughout the manuscript and would need to be substantiated further with more detailed and nuanced mechanistic investigations.

6)    Please show the BH3 mimetic data (lines 181-186) or do not include in the manuscript. Without the data to support the statements made, no conclusions can be drawn.

7)    Figure 1C – Please indicate any statistically different data points. In 21_TRAR, I expect that at least some of those points will be statistically significant. Furthermore, labels need to be shown with the y-axis. It is impossible to interpret the results from this graph with the unit values not stated.

8)    Figure 4D (line 216) – Please indicate the precise measurement for IL-8 secretion as measured by the ELISA in the text, not “about”.

9)    Figure 2B – Please perform a similar experiment but prepare protein lysates and immunoblot for TYR instead of using just qRT-PCR. Also does “transcript level” mean “expression of”. If so, please use “Relative expression” or a synonymous phrase for the axis.

10) Results – Please indicate if mycoplasma testing was performed. This is an important step for cell line authentication.

11) Figure 1A – The diagram describing the study design is overly complicated. A simple timeline would be sufficient.

12) Careful editing of the manuscript is needed. Sentence fragments are observed (lines 53-55 for example), and the paragraphs within each section can be broken into more digestible lengths. This will assist the reader in experiment and manuscript text comprehension.

Reviewer 3 Report

In this study, Koziej et al., choose two stable trametinib-resistant BRAFV600 melanoma cell lines, 29_TRAR, and 21_TRAR, and investigated alteration in phenotypes of these two melanoma cell lines in response to drug withdrawal and drug rechallenge. And found that the MITFhigh/NGFRlow/IL-8low phenotype is highly responsive to alternating periods of drug withdrawal and drug rechallenge. This is a potentially interesting study. However, there are several major points needed to be addressed: 

1, The authors only compare two cell lines, which are too small samples to get a solid conclusion. I suggest the author compare more cell lines, with at least three cell lines per group.

2, I suggest the authors overexpression or KD/KO the critical genes in melanoma cell lines to see if they can simulate corresponding phenotypes.

3, I think the authors should use clinical samples to verify the conclusions obtained.

Round 2

Reviewer 1 Report

Dear Authors,

Thank you for dealing with all my concerns.

Please check English style and grammar.

Reviewer 2 Report

Thank you to the authors for the point-by-point rebuttal to my critiques and concerns. My concerns have been adequately addressed.

Reviewer 3 Report

All the questions has been well addressed. The quality of the paper have been revised.